CERN-TH-2024-135

# Ringdown in the SYK model

**Matthew Dodelson**

*CERN, Theoretical Physics Department, CH-1211 Geneva 23, Switzerland*

ABSTRACT: We analyze thermal correlators in the Sachdev-Ye-Kitaev model away from the maximally chaotic limit. Despite the absence of a weakly curved black hole dual, the two point function decomposes into a sum over a discrete set of quasinormal modes. To compute the spectrum of modes, we analytically solve the Schwinger-Dyson equations to a high order in perturbation theory, and then numerically fit to a sum of exponentials using a technique analogous to the double cone construction. The resulting spectrum has a tree-like structure which is reminiscent of AdS black holes with curvature singularities. We present a simple toy model of stringy black holes that qualitatively reproduces some aspects of this structure.

# 1   Introduction

Consider a thermal system that is perturbed and then measured after a time $t$. At late times the system will return to equilibrium, but the nature of the approach to equilibrium is uncertain. One particularly simple behavior for the correlation function is exponential decay in time. In this situation, the decay is characterized by a complex frequency $\omega_n$ with Im $\omega_n < 0$, and the correlator behaves as $e^{-i\omega_n t}$. This is reminiscent of the final ringdown phase of black hole mergers [1], where $\omega_n$ is the lowest quasinormal mode of the final black hole.

However, exponential decay of the correlation function is by no means guaranteed. In generic systems, the approach to equilibrium is complicated by hydrodynamic effects, which lead to long-time power law tails in correlators [2, 3]. In order to disentangle such phenomena, it is useful to restrict our analysis to large $N$ systems, in which hydrodynamic tails are suppressed for generic non-conserved operators.

As an example, let us consider the two point function of glueball operators with spatial momentum $k$ in $\mathcal{N} = 4$ supersymmetric Yang-Mills theory at infinite $N$ [4].

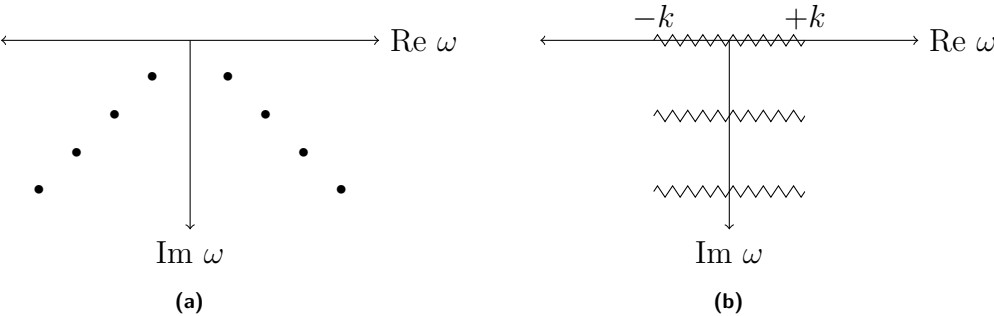

**Figure 1:** Analytic structure of the retarded Green's function at (a) $\lambda = \infty$ and (b) $\lambda = 0$.

The analytic structure of the retarded Green's function in the complex frequency plane is shown in Figure 1. At strong coupling there is a discrete spectrum of poles, corresponding to the quasinormal modes of the AdS black hole dual [5, 6]. At zero coupling there are branch cuts with real parts extending from $-k$ to $+k$. The correlator at finite coupling cannot be computed exactly, but can be analyzed with the help of various simplifying assumptions. Field-theoretic analyses tend to produce both cuts and poles at finite coupling [7, 8], while holographic models yield only poles that condense into a cut in the free limit [9, 10]. A basic tension remains between the two methods.

Since large $N$ gauge theories are difficult to solve, it is beneficial to look for examples of simpler theories where thermal correlators can be studied at finite coupling. One such theory is the Sachdev-Ye-Kitaev model [11–14]. The advantage of this system is its drastic simplification at large $N$, where the correlator satisfies an integral equation that can be readily solved. In this paper we develop a method for extracting the decay behavior of the correlator from the Schwinger-Dyson equation, extending previous work [15]. We obtain a discrete spectrum of resonances, corresponding to pure ringdown behavior of the correlator. Surprisingly, we find that several features of the holographic spectrum survive at finite coupling, including a signature of the black hole singularity [16, 17].

The plan of this paper is as follows. In Section 2 we review the moment expansion of thermal correlators. In Section 3 we explain how to compute resonance frequencies as eigenvalues of a certain non-Hermitian operator. In Section 4 we present the spectrum of resonances in SYK and discuss some of its salient features. Section 5 contains a simple bulk toy model for the SYK spectrum, and we conclude in Section 6 with some future directions.

## 2 The moment expansion

In this section we review the conjecture of [18] regarding the behavior of the moments of thermal correlators in chaotic systems. We then explain how to effectively compute the moments in the SYK model, following [18].

### 2.1 The universal operator growth hypothesis

Let us first introduce the basic objects of interest. We work at infinite temperature, and will briefly comment on the finite temperature case below. It is helpful to consider the Hilbert space spanned by operators $\mathcal{O}$, with inner product

$$(\mathcal{O}_1|\mathcal{O}_2) = \frac{\text{Tr}(\mathcal{O}_1^\dagger \mathcal{O}_2)}{\text{Tr}(1)}. \tag{2.1}$$

We also define the Liouvillian operator $\mathcal{L} = [H, \cdot]$, where $H$ is the Hamiltonian. The Wightman function and retarded Green's function of a local Hermitian operator $\mathcal{O}$ are then defined by

$$C(t) = \frac{\text{Tr}(\mathcal{O}(t)\mathcal{O}(0))}{\text{Tr}(1)} = (\mathcal{O}|e^{-i\mathcal{L}t}|\mathcal{O}), \tag{2.2}$$

$$G(t) = -\frac{i}{2}\text{Tr}(\{\mathcal{O}(t), \mathcal{O}(0)\})\theta(t) = -iC(t)\theta(t). \tag{2.3}$$

The Fourier transform of $C(t)$ gives the spectral density,

$$C(\omega) = \int_{-\infty}^{\infty} dt\, e^{i\omega t} C(t) = 2\pi(\mathcal{O}|\delta(\omega - \mathcal{L})|\mathcal{O}). \tag{2.4}$$

The retarded Green's function in frequency space is the Laplace transform of $C(t)$,

$$G(\omega) = -i \int_0^{\infty} dt\, e^{i\omega t} C(t) = \left(\mathcal{O}\left|\frac{1}{\omega - \mathcal{L} + i\epsilon}\right|\mathcal{O}\right). \tag{2.5}$$

Note that $C(t)$ is even. Assuming no singularity at the coincident point $t = 0$, we can therefore expand in even powers of $t$,

$$C(t) = \sum_{n=0}^{\infty} \mu_{2n} \frac{(it)^{2n}}{(2n)!}. \tag{2.6}$$

Here $\mu_{2n}$ are the moments of the spectral density, which can also be expressed as thermal one point functions,

$$\mu_{2n} = \int_{-\infty}^{\infty} \frac{d\omega}{2\pi} \omega^{2n} C(\omega) = (\mathcal{O}|\mathcal{L}^{2n}|\mathcal{O}). \tag{2.7}$$

The universal operator growth hypothesis [18] is a statement about the behavior of the spectral density at large real frequencies in chaotic systems, see also [19, 20]. The Wightman function is conjectured to decay exponentially,

$$C(\omega) \sim \exp\left(-\frac{\beta_0 |\omega|}{2}\right), \qquad \omega \to \pm\infty. \tag{2.8}$$

For a continuum field theory at inverse temperature $\beta$, the decay rate is given by $\beta_0 = \beta$ [21], but this is not necessarily the case for lattice systems. In particular, $\beta_0$ is nonzero at infinite temperature. Note that (2.8) is agnostic about the behavior of the spectral density in asymptotic directions away from the real axis. The asymptotic behavior away from the real axis is one of the questions we will address in this paper.

An equivalent way to state the hypothesis is in terms of the asymptotic behavior of the moments,

$$\mu_{2n} \sim \left(\frac{4n}{e\beta_0}\right)^{2n}, \qquad n \to \infty. \tag{2.9}$$

It follows that the series (2.6) is convergent around the origin, with radius of convergence $|t| = \beta_0/2$. Given the moments up to $\mu_{2n}$, we can therefore compute the correlation function at a time $0 < t < \beta_0/2$ up to an error $(2t/\beta_0)^{2n}$ which is exponentially small in $n$. For chaotic spin systems the computation of the moments is limited by the exponential growth of complexity of the operators $\mathcal{L}^{2n}\mathcal{O}$, and the record number of computed moments is 45 [22]. In the SYK model one can do far better, as we will review next.

## 2.2 Moments in the SYK model

The SYK model is a quantum mechanical system with $N$ fermions, with a random interaction term that couples an even number $q$ fermions [11–14]. The Hamiltonian is

$$H = i^{q/2} \sum_{1 \leq i_1 < \ldots < i_q \leq N} j_{i_1 \ldots i_q} \psi_{i_1} \cdots \psi_{i_q}, \qquad \langle j_{i_1 \ldots i_q}^2 \rangle = \frac{(q-1)! \mathcal{J}^2}{2^{1-q} q N^{q-1}}. \tag{2.10}$$

Here we have chosen a convenient normalization for the dimensionful coupling $\mathcal{J}$.

We are interested in the infinite $N$ limit, in which the entropy becomes infinite and correlation functions are expected to decay to zero at late times. In this limit, the two point function of an elementary fermion $\mathcal{O} = \sqrt{2}\psi_1$ at infinite temperature satisfies the Schwinger-Dyson equation [11, 15, 18],

$$\omega G(\omega) = 1 + \frac{2\mathcal{J}^2}{q} G(\omega)\Sigma(\omega), \tag{2.11}$$

$$\Sigma(\omega) = -i \int_0^\infty dt \, e^{i\omega t} C(t)^{q-1}. \tag{2.12}$$

Here $\omega$ should be evaluated slightly above the real axis. These equations can be solved perturbatively in $\mathcal{J}$ using a recursive algorithm [18], which we repeat here for convenience:

1. Set $G_0(\omega) = \omega^{-1}$.

2. Compute $C_j(t)$ from $G_j(\omega)$ by replacing $\omega^{-2n-1}$ with $(it)^{2n}/(2n)!$.

3. Compute $\Sigma_j(\omega)$ by expanding $C_j(t)^{q-1}$ to order $t^j$ and then replacing $(it)^{2n}$ with $\omega^{-2n-1}(2n)!$.

4. Set $G_{j+1}(\omega) = (1 + 2q^{-1}\mathcal{J}^2 G_j(\omega)\Sigma_j(\omega))/\omega$ up to order $\omega^{-2j-1}$.

Once $G_j(\omega)$ has been computed using this protocol, the moments can be read off of $G_j(\omega)$ via the relation

$$G_j(\omega) = \sum_{k=0}^{j} \frac{\mu_{2k}}{\omega^{2k+1}}. \tag{2.13}$$

The moments are integers when $\mathcal{J}$ is normalized appropriately. The crucial point is that the above procedure for computing $\mu_{2j}$ scales polynomially in time with $j$. By running this algorithm for several weeks on a computer, we were able to obtain 2000 moments for the case $q = 4$, and 1500 moments for several other values of $q$. We have included the results in a supplementary file.

At this point the reader may be questioning the utility of these results. The first 50 moments can be generated in a matter of seconds, and are sufficient for determining the correlation function up to $t = \beta_0/4$ to 30 digits of precision. Naively it seems like this should be enough precision for any reasonable purpose.

The reason we found it necessary to push further is two-fold. First, we would like to compute resonance frequencies $\omega_n$ with large imaginary part. These contribute to the correlation function with magnitude $e^{-\text{Im}(\omega_n)t}$, which is very small even when $t$ is order one. Second, we need to reconstruct the resonances from the moment expansion, which is only convergent until $t = \beta_0/2$. At $t = \beta_0/2$ the correlator has generally not dropped far below its initial value, so the onset of exponential decay is difficult to see from the moment expansion. As we will see in the next section, it is indeed possible to fit a signal over a short period of time to a sum of exponentials, but an exceptional amount of precision is required. To this end, it is helpful to compute as many moments as possible.

Let us briefly comment on the generalization to finite temperature [23]. In this case there is a dimensionless parameter $\beta\mathcal{J}$, and it is possible to systematically compute

the moments as a power series in $\beta\mathcal{J}$. We have not studied the convergence properties of this series or the additional computational overhead, and we will restrict to infinite temperature in this paper.

## 3   Fitting to exponentials

In the previous section we saw how to compute the function $C(t)$ to arbitrary precision inside its radius of convergence $|t| < \beta_0/2$. Our task is now to fit this function to a sum of exponentials,

$$\sum_{n=0}^{\infty} \mu_{2n} \frac{(it)^{2n}}{(2n)!} = C(t) = \sum_{n=0}^{\infty} d_n e^{-i\omega_n t}, \qquad 0 \le t < \frac{\beta_0}{2}. \tag{3.1}$$

Since the system thermalizes at late times, we expect that $\mathrm{Im}\, \omega_n < 0$. Note that the left hand side of (3.1) is simply the perturbative expansion of $C(t)$, since $\mu_{2n} \sim \mathcal{J}^{2n}$ at infinite temperature. By dimensional analysis we have $\omega_n \sim \mathcal{J}$, so the right hand side of (3.1) is a nonperturbative resummation of the moment expansion. Moreover, the sum on the right hand side converges for all $t \ge 0$.

Fortunately, it is not necessary to perform the resummation at the level of Feynman diagrams. In this section we will review how to solve the fitting problem (3.1) by diagonalizing a large matrix [24]. We will then explain how this diagonalization procedure is related to other computations, including the double cone construction in holography [25].

### 3.1   Exponential fitting as an eigenvalue problem

Following [24], our strategy for solving (3.1) is to truncate the sum on the right hand side at $n = S - 1$. We then solve (3.1) on a finite grid with size $2S$ and spacing $\Delta t$. This yields $2S$ equations for $2S$ unknowns $d_n$ and $\omega_n$,

$$C(j\Delta t) = \sum_{n=0}^{S-1} d_n e^{-i\omega_n j\Delta t}, \qquad j = 0, \ldots, 2S - 1. \tag{3.2}$$

We will eventually take the limit $S \to \infty$ while holding $S\Delta t$ fixed.

The equations (3.2) look nonlinear, but in fact they can be solved using linear algebra. To see this, let us suppose that there exists an operator $\tilde{U}_S$ with $S$ eigenvalues, which satisfies the relation

$$C(j\Delta t) = (\mathcal{O}|\tilde{U}_S^j|\mathcal{O}), \qquad j = 0, \ldots, 2S - 1. \tag{3.3}$$

Comparing (3.2) with (3.3), we see that the eigenvalues of $\tilde{U}_S$ can be identified with the complex frequencies $\omega_n$,

$$\tilde{U}_S|\omega_n) = e^{-i\omega_n \Delta t}|\omega_n). \tag{3.4}$$

The problem of computing the resonances is therefore reduced to finding an operator $\tilde{U}_S$ satisfying (3.3). Note that the eigenvalues of $\tilde{U}_S$ do not lie on the unit circle. Therefore $\tilde{U}_S$ is not unitary, in contrast to the usual time evolution operator $U = e^{-i\mathcal{L}\Delta t}$.

To construct $\tilde{U}_S$, we consider the time-evolved states

$$|j) = \tilde{U}_S^j|\mathcal{O}). \tag{3.5}$$

The discrete Krylov subspace is defined as the span of the vectors $|j)$ for $j = 0, 1, \ldots S-1$. We now make an ansatz where the right eigenvectors of $\tilde{U}_S$ are in the Krylov subspace, and can therefore be expanded as

$$|\omega_n) = \sum_{j=0}^{S-1} B_{nj}|j). \tag{3.6}$$

Taking the inner product of (3.6) with $(0|\tilde{U}_S^k$ and plugging into (3.4), we find the generalized eigenvalue equation

$$\sum_{j=0}^{S-1} \left( C((j+k+1)\Delta t) - C((j+k)\Delta t)e^{-i\omega_n \Delta t} \right) B_{nj} = 0. \tag{3.7}$$

After solving for the frequencies $\omega_n$, we are left with a system of linear equations (3.2) for $d_n$ which can be readily solved.

At this point we would like to mention that there is nothing sacrosanct about the Krylov basis. In applications to quantum chemistry, one is usually most interested in long-lived resonances, and in this setting a Fourier basis is more natural [24, 26]. In the SYK model we do not have any reason to expect long-lived excitations, and the Krylov basis will suffice for our purposes.

## 3.2 Removing the cutoff

Let us now consider the limit $S \to \infty$. Notice that the eigenvalue equation (3.7) depends on the correlation function at discrete time steps from $t = 0$ up until $t_{\max} = (2S-1)\Delta t$. Since the moment expansion only converges for $|t| < \beta_0/2$, we want to take $t_{\max} < \beta_0/2$ as in Figure 2, with $t_{\max}$ fixed as a function of $S$. It follows that the grid spacing $\Delta t$

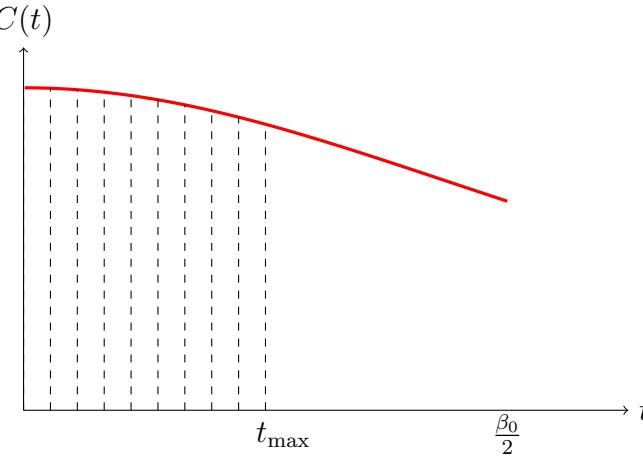

**Figure 2:** The dashed lines mark the discrete sampling times that enter the eigenvalue equation (3.7) on a grid with $S = 5$. The moment expansion gives the correlator up until $t = \beta_0/2$, and the largest time sampled is $t_{\text{max}} < \beta_0/2$.

goes to zero in the limit $S \to \infty$. In this limit, it is natural to consider the following non-Hermitian operator,

$$\tilde{\mathcal{L}} = \lim_{S \to \infty} \frac{i}{\Delta t} \log \tilde{U}_S. \tag{3.8}$$

The spectrum of the operator $\tilde{\mathcal{L}}$ is potentially subtle, since the eigenvalues of $\tilde{U}_S$ could accumulate in the limit $S \to \infty$. Assuming that the spectrum of $\tilde{\mathcal{L}}$ is discrete and using (3.1), the retarded Green's function becomes

$$\left( \mathcal{O} \left| \frac{1}{\omega - \mathcal{L}} \right| \mathcal{O} \right) = G(\omega) = \sum_{n=0}^{\infty} \frac{d_n}{\omega - \omega_n} = \left( \mathcal{O} \left| \frac{1}{\omega - \tilde{\mathcal{L}}} \right| \mathcal{O} \right). \tag{3.9}$$

It follows that $G(\omega)$ is a meromorphic function with poles in the lower half plane. In contrast, if the eigenvalues of $\tilde{\mathcal{L}}$ are not discrete, then $G(\omega)$ contains branch cuts (or even worse types of singularities). In theories with a continuous spectrum but a finite number of local degrees of freedom, we expect branch cuts due to hydrodynamic long time tails [2, 3]. In a large $N$ theory it is an open question whether $\tilde{\mathcal{L}}$ has a discrete spectrum. Note that the representations of the Green's function on the left-hand and right-hand side of (3.9) look superficially similar, but they are quite different since $\mathcal{L}$ always has a continuous spectrum in the thermodynamic limit.

The limit $S \to \infty$ has an interesting bulk avatar when the theory is holographic. In [25], it was shown that the ramp in the spectral form factor can be reproduced by the double cone geometry, which is an AdS black hole periodically identified in time under $t \to t + T$. Naively, the partition function of this geometry is computed by the trace

$\text{Tr}(e^{-iKT})$, where $K$ is the generator of boosts. However, the definition of $K$ requires an $i\epsilon$ prescription due to the degenerating time circle at the horizon. Therefore one should actually compute $\text{Tr}(e^{-i\tilde{K}T})$, where $\tilde{K}$ is a modified non-Hermitian boost operator. It was shown in [27, 28] that the eigenvalues of $\tilde{K}$ in the limit $\epsilon \to 0$ are precisely the quasinormal mode frequencies. The boundary analog of this statement is that after taking $S \to \infty$, the eigenvalues of $\tilde{\mathcal{L}}$ are the frequencies $\omega_n$.

We conclude this section by mentioning the connection to classical chaos. In the context of dynamical systems, one can consider the correlation function $C(t)$ of a function on phase space. Here $t$ can be either a discrete or continuous variable. When the late-time decay of $C(t)$ to equilibrium takes an exponential form, the complex frequencies that characterize the decay are known as Pollicott-Ruelle resonances [29–31]. These resonances can be computed as eigenvalues of a modified time evolution operator [32, 33], in direct analogy with our analysis. For generic systems the spectrum must be analyzed numerically [34–36]. The quantum setup that we have discussed is much less explored, but see [37, 38] for one example.

## 4   Resonances in SYK

In this section we apply the methodology developed earlier in the paper to the specific case of the SYK model. We first present the $q = 4$ answer, and comment on the similarities to the spectrum of AdS black holes with curvature singularities. We then discuss the large $q$ limit, in which we reproduce known results [39].

### 4.1   The $q = 4$ tree

Let us start by solving the generalized eigenvalue equation (3.7) in the case $q = 4$. The maximum grid size that can be used is limited by the number of available moments. We use the following protocol to search for modes.

1. Pick a grid size $S$ and number of moments $n_{\text{max}}$.

2. Solve the generalized eigenvalue equation (3.7) with $n_{\text{max}}$ moments and $t_{\text{max}} \sim \beta_0/4$ for the $S$ frequencies $\omega_n$.

3. Increase $n_{\text{max}}$ until convergence is obtained for the spectrum.

4. Repeat the computation with a grid of size $.9S$ and compare the spectra. Modes that agree between the two grids within a tolerance of $.1$ are said to have converged.

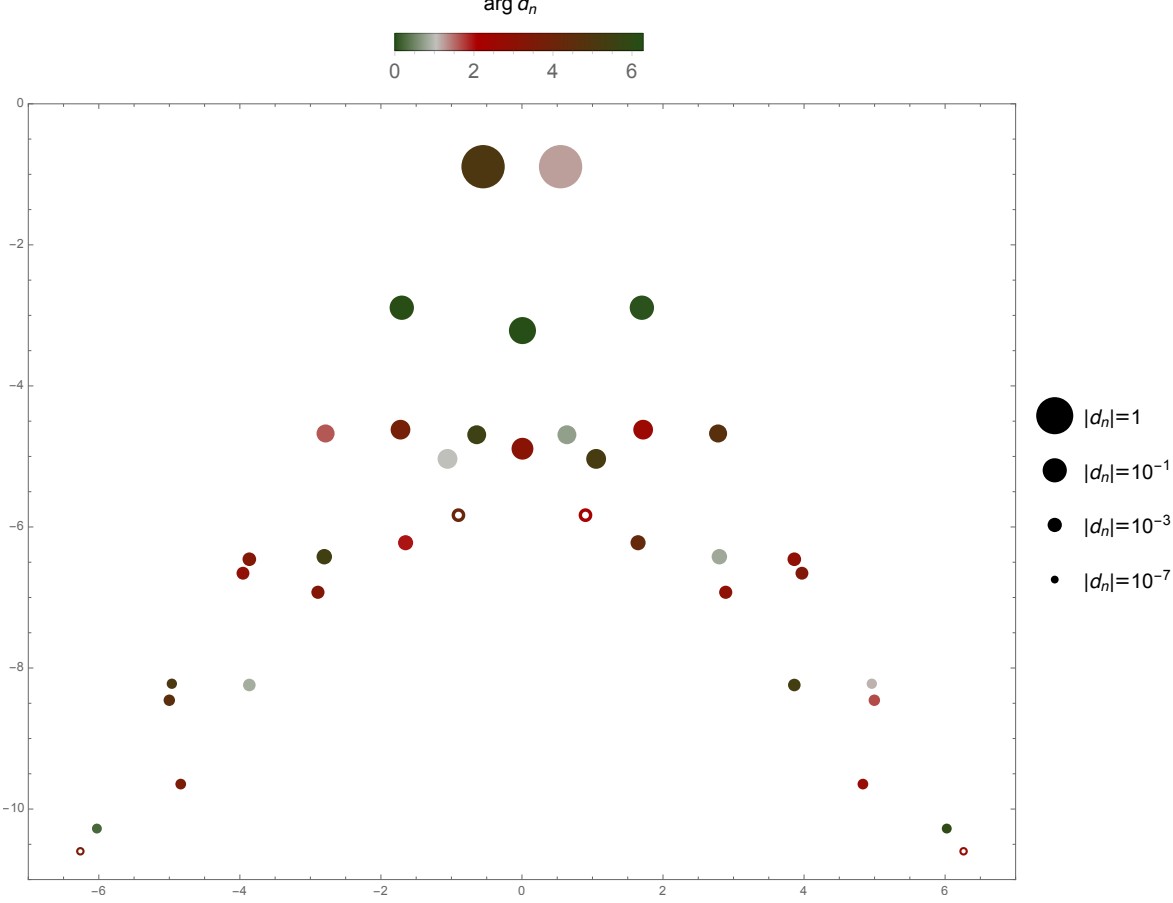

**Figure 3:** The spectrum of resonances in $q = 4$ SYK. The magnitude and phase of the residues $d_n$ are indicated by the size and color of the dots, as specified in the legends. The solid dots have converged according to the criterion put forth in the main text, whereas the hollow dots have not yet converged.

The choice $t_{\mathrm{max}} \sim \beta_0/4$ is somewhat arbitrary, and any $t_{\mathrm{max}} < \beta_0/2$ should give the same results in the $S \to \infty$ limit. Following this procedure with 2000 moments leads to a maximum allowed grid size of around $S = 670$. We have included the complete set of converged modes in the supplementary files, and the results are shown in Figure 3 (from now on we set $\mathcal{J} = 1$).

Note that the imaginary part of the slowest mode agrees with the result derived in [15]. In fact, this mode has already converged to three digits with 50 moments and a grid size of 20. We have observed that the modes with smallest imaginary part tend to converge the fastest, and we expect that increasing the grid size will add more poles in the lower half of Figure 3.

We would now like to point out several similarities between Figure 3 and the spectrum of quasinormal modes of an AdS black hole [5, 6].

| $\omega_n$ | $d_n$ |
|---|---|
| $\pm.54 - .89i$ $(+38)$ | $.4 \pm 1.2i$ $(+38)$ |
| $\pm1.7 - 2.9i$ $(+22)$ | $.050 \pm .002i$ $(+20)$ |
| $-3.2i$ $(+18)$ | $.11$ $(+16)$ |
| $\pm1.7 - 4.6i$ $(+8)$ | $-.0075 \pm .0053i$ $(+5)$ |
| $\pm2.8 - 4.7i$ $(+12)$ | $-.0002 \mp .0040i$ $(+9)$ |
| $\pm.6 - 4.7i$ $(+4)$ | $.0046 \pm .0041i$ $(+1)$ |
| $-4.9i$ $(+2)$ | $-.022$ $(+2)$ |
| $\pm1.1 - 5.0$ $(+3)$ | $.0049 \mp .0083i$ |

(a) This table contains the first 14 frequencies for $q = 4$, with the additional number of converged digits indicated to the right of the frequency. The $d_n$'s have converged as well, albeit to a slightly lower level of precision.

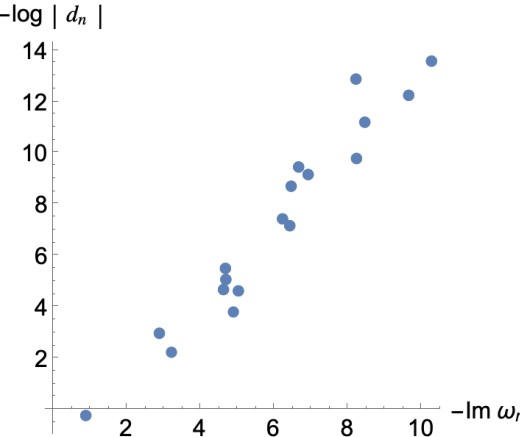

(b) Here we display the linear correlation between $\log |d_n|$ and Im $\omega_n$. In this plot we have included all the converged modes for $q = 4$.

**Figure 4**

- **Meromorphy:** Recall that in the holographic limit, the Green's function is meromorphic in the complex frequency plane. This is a general consequence of the properties of the wave equation with asymptotically AdS boundary conditions in the presence of a nonextremal horizon [4, 17, 40–42]. In our method, branch points would appear as accumulation points of poles in the limit $S \to \infty$. We do not see any obvious sign of accumulating poles in our numerics. In particular, for $|\text{Im } \omega_n| < 5$ we find a discrete spectrum of modes that seem to have converged to at least four digits, see Figure 4a. However, we would like to emphasize that it is not possible to prove meromorphy using our numerical method, and it remains a logical possibility that a cut forms as the grid size is increased further.

- **Tree structure:** The asymptotic structure of quasinormal modes in AdS black holes is well understood. In the case of AdS$_{d+1}$-Schwarzschild branes, the modes are asymptotically equally spaced, at an angle $\pi/d$ from the real axis [43, 44]. For $d > 2$ the asymptotic angle is less than $\pi/2$ and the picture resembles a tree (see Figure 1a). For more general black holes there can be a more complicated asymptotic structure of modes [10, 42], but the spectrum is still contained within an angular sector $\pi + \theta < \arg \omega < 2\pi - \theta$ for some angle $\theta$. We find such a tree structure in Figure 3, with $\theta \sim 1.04$.[1] Our expectation is that pushing the

---

[1]This angle is close to $\pi/3$, which is the asymptotic angle of frequencies in an AdS$_4$ black brane. We think this is probably a coincidence.

computation further would populate the angular sector with infinitely many more poles with increasing $|\text{Im } \omega_n|$.

- **Exponential decay at $\omega = \pm i\infty$:** In [17, 41] it was argued that the spectral density in black holes with curvature singularities decays exponentially at large imaginary frequency. This exponential decay is a signature of null geodesics that bounce off the singularity [16, 45–47]. A related property that is more easily verified in our setup is the exponential decay of the residues of the poles as a function of $\text{Im } \omega_n$ [42]. We found that the relation between $\log|d_n|$ and $\text{Im } \omega_n$ is indeed approximately linear, see Figure 4b. This can also be seen in Figure 3, which employs a logarithmically varying scale for $|d_n|$.

Since we are far away from the holographic limit, the bulk is highly stringy (by which we mean that higher derivative corrections are not suppressed). It is therefore a bit unexpected that these three properties hold in our context. In Section 5 we present a toy model of stringy black holes that qualitatively captures all three features.

## 4.2  Transition to large $q$

Next we briefly discuss the limit $q \to \infty$. The work [39] presented numerical evidence that the correlator exponentiates in this limit,

$$C(t) \sim \frac{1}{(\cosh t)^{2\Delta}}, \qquad \Delta = \frac{1}{q} \to 0. \tag{4.1}$$

Here we have set $\mathcal{J} = 1$ and $\beta = 0$. The resonance data is

$$\omega_n = -2i(\Delta + n), \tag{4.2}$$

$$d_n = \frac{(-1)^n}{n!} \frac{2^{2\Delta}\Gamma(2\Delta + n)}{\Gamma(2\Delta)}, \tag{4.3}$$

with $n \geq 0$. Note that the frequencies $\omega_n$ are all imaginary, and that the residues $d_n$ behave like a power law in $n$ for large $n$. These features are suggestive of a dual black hole description with no curvature singularity (such as a BTZ black hole [48]).

In Figure 5 we display the results for several values of $q$ up to $q = 20$. Here the eigenvalue equation was solved with 1500 moments. We find that the opening angle of the tree decreases with increasing $q$, and the poles eventually collapse onto the imaginary axis. For $q = 20$ all the poles up to $|\text{Im } \omega_n| = 11$ are imaginary. Moreover, the poles start to form clusters with residues of alternating sign around the frequencies (4.2). We expect that as $q$ is increased further, the width of the clusters shrinks to zero, reproducing the result (4.2). Further numerical experiments confirm this expectation.

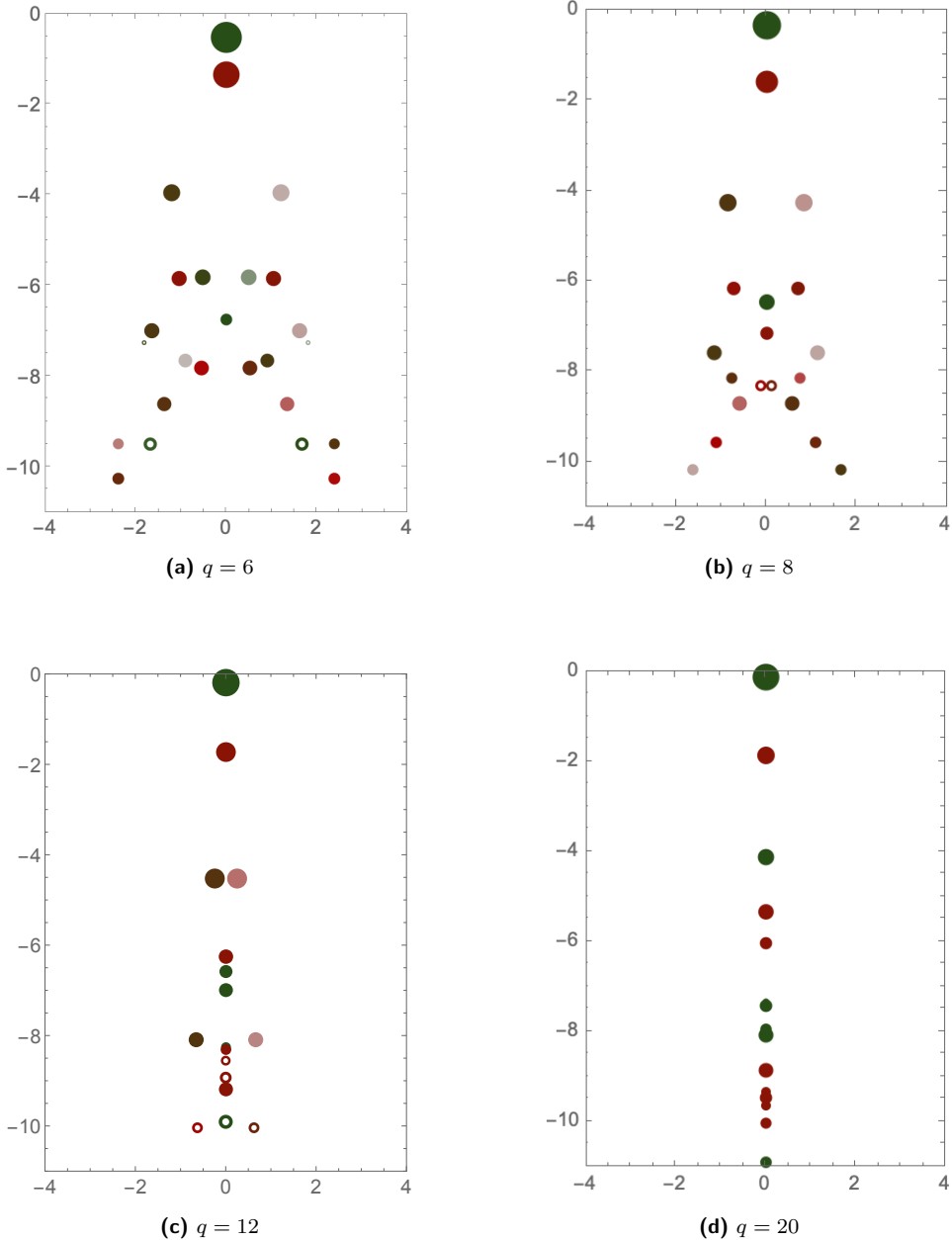

**Figure 5:** The transition from order one $q$ to $q = \infty$. The legends are identical to those in Figure 3.

# 5 Stringy black holes

In this section we present a crude holographic model for the spectrum of resonances in finite $q$ SYK. The model consists of a single light field propagating on a black hole background. Since we are in the stringy regime we expect that such a naive description should not be able to capture the full answer. After studying the model we will explain

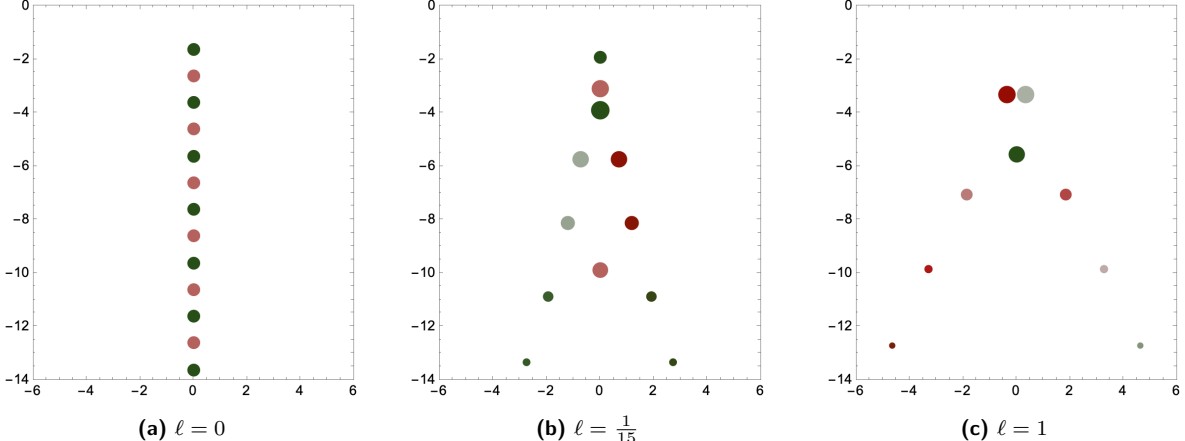

**Figure 6:** The quasinormal modes for the model (5.2) in the background $f(r) = r^2 - 1 - \ell/r^2 + \ell/(4r^4)$ for various values of $\ell$. At $\ell = 0$ the residues grow too fast to be plotted in a useful way, so we choose a constant point size instead.

its limitations.

Let us start with the large $q$ limit. In this regime there is an emergent $\text{AdS}_2$ dual to the SYK model at finite coupling, as described in [49]. We consider $\text{AdS}_2$ in Schwarzschild-like coordinates,

$$ds^2 = -f(r)\,dt^2 + \frac{dr^2}{f(r)}, \qquad f(r) = r^2 - 1. \tag{5.1}$$

Here we have set the AdS radius to one. Our toy model contains a scalar field $\phi$ with mass $m = 1$ satisfying the free wave equation,

$$(\Box - 1)\phi = 0. \tag{5.2}$$

The resulting quasinormal modes are shown in Figure 6a. The modes are equally spaced and purely imaginary, as in the large $q$ result (4.2). Strictly speaking, it would be more applicable to consider a fermionic operator with dimension $\Delta = 1/q$, but a massive scalar will suffice for our qualitative picture.

Away from the large $q$ limit, we saw in Figure 5 that the spectrum opens up into a tree. In our toy model we aim to replicate this structure by introducing a black hole singularity at $r = 0$. To do so, we perturb the redshift factor as

$$f(r) = r^2 - 1 + \sum_{n=1}^{\infty} \frac{f_n}{r^n}, \tag{5.3}$$

where the parameters $f_n$ are small in the large $q$ limit. We calculate the quasinormal modes using the Mathematica package `QNMSpectral` [50], and we compute the residues

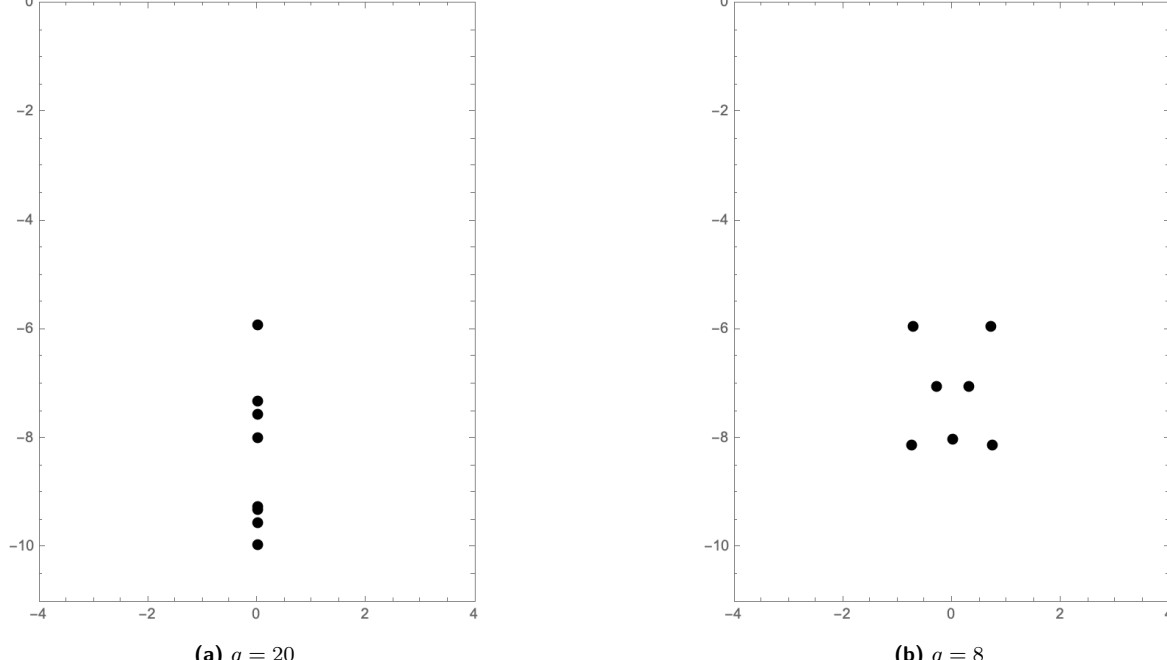

**(a)** $q = 20$                **(b)** $q = 8$

**Figure 7:** Converged zeroes of the spectral density for two different values of $q$.

$d_n$ using the thermal product formula [42]. As $f_n$ is increased, a tree generically forms with modes inside, as shown in Figures 6b and 6c for a specific choice of the $f_n$ coefficients. This mimics the transition from large $q$ to finite $q$ that we found in Section 4.2.

The most glaring difference between Figure 6 and the result in finite $q$ SYK is in the density of modes. Let us define $N(\Lambda)$ as the number of modes with $-\Lambda < \text{Im } \omega_n < 0$. In our toy model, $N(\Lambda)$ quickly approaches linear growth, so that $N(\Lambda) \propto \Lambda$ for large $\Lambda$. This linear growth was argued in [42] to be a general feature of the free wave equation on an AdS black hole background. In such a spacetime, the boundary Wightman function can be shown to have no zeroes in the complex frequency plane [17, 41]. Combining the no-zeroes property with the exponential decay of the Wightman function at $\omega = \pm\infty$ leads to linear growth of $N(\Lambda)$ at large $\Lambda$. This linear behavior will always be present in our toy model, independently of the specific choice of $f(r)$.

Now let us discuss $N(\Lambda)$ in finite $q$ SYK. Referring to Figures 3 and 5, we see that $N(\Lambda)$ seems to grow faster than linearly with $\Lambda$. This suggests the presence of zeroes in the spectral density, which solve the equation

$$C(\omega) = \sum_{n=0}^{\infty} \frac{2id_n\omega_n}{\omega^2 - \omega_n^2} = 0. \tag{5.4}$$

We solve this equation by placing an upper cutoff on the sum and looking for solutions that converge as the cutoff is increased. The results are plotted in Figure 7. Note that the zeroes are interlaced with the poles in the large $q$ limit, so that a single pole at $q = \infty$ splits into a cluster of zeroes and poles at large but finite $q$.

Recall that when $q$ is strictly infinite, there are no zeroes in $C(\omega)$ [42]. Models satisfying this condition often appear in the time series literature, where they are referred to as autoregressive [51]. The no-zero property is also true in the large $q$ SYK chain [52], and in [42] it was suggested that this property holds more generally. Here we see that no-zeroes is violated at finite $q$. It would be interesting to find a physical interpretation for zeroes in the spectral density.

Since the toy model we have presented ultimately fails, it is natural to ask whether one can construct a more accurate model. For instance, one might consider higher order wave equations, which are expected to arise from higher derivative terms in the bulk action. A true bulk description of SYK at finite $q$ and nonzero temperature may give clues into the nature of the black hole singularity at finite string length, and we leave this as an open question.

## 6    Discussion

In this work we showed that the SYK model at infinite temperature shares certain surprising features with AdS black holes. Our flagship result is Figure 3, which displays a discrete spectrum of modes arranged in a tree, with exponentially decaying residues.

We conclude with some future directions.

- It would be interesting to search for some underlying universality that governs the statistics of the spectra we have found, analogous to random matrix theory. This question was recently analyzed in the related context of open systems [53, 54]. We would also like to understand why our spectra do not contain continuous sectors, which would appear as branch cuts in the complex frequency plane. Perhaps this is related to the quantum Anosov condition [55, 56].

- This paper has addressed the spectrum of modes but not the eigenstates themselves. In chaotic systems the eigenstates often take a fractal form [37, 38, 57]. Maybe the same is true in the SYK model.

- A natural step toward the eventual goal of understanding thermalization in large $N$ gauge theories is to repeat our analysis in large $N$ matrix quantum mechanics [58]. One particularly important theory is the BFSS model [59–61], in which the quasinormal modes at strong coupling form a tree [62]. We think it might be

possible to compute the resonances at finite coupling using the moment method, with the help of the matrix bootstrap [63, 64].

- Since the SYK model is 0+1 dimensional, it does not yield much insight into intrinsically higher dimensional features of black holes. One such phenomenon is the photon sphere, which is probed by the bulk cone limit on the boundary [65, 66]. The bulk cone singularity is expected to be resolved into a bump by stringy effects [67]. It would be interesting to see if this expectation is borne out in higher dimensional generalizations of SYK [68, 69].

## Acknowledgments

We thank Y. Chen, A. Dymarsky, E. Gesteau, A. Grassi, T. Hartman, C. Iossa, R. Karlsson, S. Komatsu, N. Lashkari, H. Lin, H. Liu, R. Mahajan, J. Maldacena, P. Nayak, K. Papadodimas, M. Rangamani, S. Shenker, J. Sonner, D. Stanford, and A. Zhiboedov for inspiration. We are grateful to A. Grassi, R. Karlsson, and A. Zhiboedov for comments on the draft. This project has received funding from the European Research Council (ERC) under the European Union's Horizon 2020 research and innovation programme (grant agreement number 949077). This research was supported in part by grant NSF PHY-2309135 to the Kavli Institute for Theoretical Physics (KITP).

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
