# Peer review of "Ringdown in the SYK model"

_SciPost Physics_

## Round 2 · Referee Report · Anonymous (Referee 1) · 2025-4-21

Strengths
Weaknesses
Report
The main result is that the spectrum remains meromorphic, which is a very important statement. To some, this may have been expected and to some, a surprise. Regardless, the answer seems very clear and will provide a useful and invaluable benchmark in the future. Hence, the paper certainly eventually deserves to be published by SciPost. In order to make the presentation clearer, below, I suggest a number of small improvements to the text.
Requested changes
-
It would be nice to explain more about the SYK model itself. For example, why we think of the maximal chaos limit as the `strongly coupled' limit, or, more accurately, as analogous to the large-N limit of higher dimensional models.
-
I'm not sure that I like the aesthetics of the figures (e.g., the sizes of dots depending on the residue), but I do leave this up to the author to decide. Also, $d_n$ could be more clearly defined in Section 4 so that one doesn't have to search through the previous sections.
-
Is there any sense in which one could claim that there are several branches of QNMs visible in the spectrum similar to the holographic calculation in [10]? Figure 3 kind of looks like that but it's hard to say.
-
I think it would be very interesting to plot the spectral functions as a functions of real omega. I'm thinking of this along the lines of
- https://arxiv.org/abs/hep-th/0602059
- https://arxiv.org/abs/hep-ph/0602044
- https://arxiv.org/abs/1806.10997 Spectral functions themselves can teach us a lot about the transition of the physical spectrum from strong to weak coupling.
Recommendation
Ask for minor revision

Author: Matthew Dodelson on 2025-08-07 [id 5704]
(in reply to Report 1 on 2025-04-21)Thank you for the helpful comments and suggestions. I made the requested changes in the new version. Regarding point 3, it is not so clear to me whether the branch structure is appearing here. In particular there are clusters of QNMs in SYK which are very close together, and there is no analog of this in holographic models. So I cited [10], but prefer not to speculate further.

---

## Round 2 · Referee Report · Anonymous (Referee 2) · 2025-4-25

Strengths
-Clever use of the Krylov basis
-Interpolation away and towards the holographic limit
Weaknesses
Report
I think this paper should be published subject to an explanation of how one estimates the radius of convergence $\beta_0$. I could not find this explained in the manuscript, despite looking through it several times. Perhaps I missed something.
Requested changes
Please explain more prominently how $\beta_0$ is determined for this model.
Recommendation
Publish (surpasses expectations and criteria for this Journal; among top 10%)

---

## Editorial Decision

unknown